# An Insight into the microRNAs Associated with Arteriovenous and Cavernous Malformations of the Brain

**DOI:** 10.3390/cells10061373

**Published:** 2021-06-02

**Authors:** Ioan Alexandru Florian, Andrei Buruiana, Teodora Larisa Timis, Sergiu Susman, Ioan Stefan Florian, Adrian Balasa, Ioana Berindan-Neagoe

**Affiliations:** 1Clinic of Neurosurgery, Cluj County Emergency Clinical Hospital, 400012 Cluj-Napoca, Romania; stefanfloriannch@gmail.com; 2Department of Neurosurgery, Iuliu Hatieganu University of Medicine and Pharmacy, 400012 Cluj-Napoca, Romania; 3Department of Medical Genetics, Iuliu Hațieganu University of Medicine and Pharmacy, 400012 Cluj-Napoca, Romania; andrei.buruiana@yahoo.com; 4Department of Physiology, Iuliu Hatieganu University of Medicine and Pharmacy, 400006 Cluj-Napoca, Romania; timis.teodora@umfcluj.ro; 5Department of Morphological Sciences—Histology, Iuliu Hatieganu, University of Medicine and Pharmacy, 400012 Cluj-Napoca, Romania; sergiu.susman@umfcluj.ro; 6Clinic of Neurosurgery, Tîrgu Mureș County Clinical Emergency Hospital, 540136 Tîrgu Mureș, Romania; adrian.balasa@yahoo.fr; 7Department of Neurosurgery, Tîrgu Mureș University of Medicine, Pharmacy, Science and Technology, 540139 Tîrgu Mureș, Romania; 8Research Center for Functional Genomics, Biomedicine and Translational Medicine, Institute of Doctoral Studies, “Iuliu Hatieganu” University of Medicine and Pharmacy, 400012 Cluj-Napoca, Romania; ioana.neagoe@umcluj.ro

**Keywords:** brain arteriovenous malformations, cerebral cavernous malformations, microRNA

## Abstract

**Background**: Brain arteriovenous malformations (BAVMs) and cerebral cavernous malformations (CCMs) are rare developmental anomalies of the intracranial vasculature, with an irregular tendency to rupture, and as of yet incompletely deciphered pathophysiology. Because of their variety in location, morphology, and size, as well as unpredictable natural history, they represent a management challenge. MicroRNAs (miRNAs) are strands of non-coding RNA of around 20 nucleotides that are able to modulate the expression of target genes by binding completely or partially to their respective complementary sequences. Recent breakthroughs have been made on elucidating their contribution to BAVM and CCM occurrence, growth, and evolution; however, there are still countless gaps in our understanding of the mechanisms involved. **Methods**: We have searched the Medline (PubMed; PubMed Central) database for pertinent articles on miRNAs and their putative implications in BAVMs and CCMs. To this purpose, we employed various permutations of the terms and idioms: ‘arteriovenous malformation’, ‘AVM’, and ‘BAVM’, or ‘cavernous malformation’, ‘cavernoma’, and ‘cavernous angioma’ on the one hand; and ‘microRNA’, ‘miRNA’, and ‘miR’ on the other. Using cross-reference search; we then investigated additional articles concerning the individual miRNAs identified in other cerebral diseases. **Results**: Seven miRNAs were discovered to play a role in BAVMs, three of which were downregulated (miR-18a, miR-137, and miR-195*) and four upregulated (miR-7-5p, miR-199a-5p, miR-200b-3p, and let-7b-3p). Similarly, eight miRNAs were identified in CCM in humans and experimental animal models, two being upregulated (miR-27a and mmu-miR-3472a), and six downregulated (miR-125a, miR-361-5p, miR-370-3p, miR-181a-2-3p, miR-95-3p, and let-7b-3p). **Conclusions**: The following literature review endeavored to address the recent discoveries related to the various implications of miRNAs in the formation and growth of BAVMs and CCMs. Additionally, by presenting other cerebral pathologies correlated with these miRNAs, it aimed to emphasize the potential directions of upcoming research and biological therapies.

## 1. Introduction

Brain arteriovenous malformations (BAVMs) are predominantly congenital vascular disorders that may arise anywhere inside of the central nervous system [1,2,3,4,5,6]. They are comprised of one or more arterial feeders supplying a vascular nidus, and one or more draining veins. The nidus itself represents the site where arterial blood is shunted directly into the venous system without an interpolating network of capillaries. As a result, the blood flows at a high pressure through these lesions, making them susceptible to hemorrhagic rupture, this being their most common form of presentation. A small number have also been described as appearing later in life after a different cerebral injury (trauma, infection, stroke, tumors etc.), making their pathophysiology and genetic determination all the more intriguing and poorly understood [7,8].

Cerebral cavernous malformations (CCMs), or cavernomas, are vascular lesions that, unlike BAVMs, are generally small and well delineated, with a low blood flow within them [9,10,11,12,13]. Aside from the majority of them being congenital, they have also been observed to appear in patients subjected to radiosurgery [14,15,16]. They are mostly asymptomatic and discovered incidentally on imaging studies; however, they can present via epilepsy or hemorrhagic stroke. The sporadic occurrence of CCMs typically denotes a single isolated lesion possibly linked to a developmental venous anomaly (DVA), whereas the familial variety leads to the formation of multiple lesions spread diffusely within the brain. Familial forms are correlated with an autosomal dominant mutation in any of the three key CCM genes (CCM1/KRIT1, CCM2/OSM, or CCM3/PDCD10) [10]. Both BAVMs and CCMs can be effectively treated via microsurgical resection, ensuring symptomatic control and prevention of any future hemorrhage occurring from these lesions.

MicroRNAs (miRNA or miR) represent short strands of non-coding ribonucleic acid (RNA), approximately 20–22 nucleotides long, with the capacity to regulate the expression of target genes via the complete or partial binding onto their respective complementary sequences [17,18,19,20,21,22]. As a result, they hamper translation while also stimulating mRNA degradation. Currently, there is a shortage of data concerning the multitude of roles miRNAs play in the pathophysiology of AVMs and CCMs. In the following, we will present a summary of the miRNAs discovered to be dysregulated in these vascular pathologies, along with the pathways affected in other.

## 2. Methods

The purpose of this review is to offer an illustrative synopsis on the recent findings related to miRNAs associated with BAVMs and CCMs. We explored the electronic databases of Medline (PubMed, PubMed Central) by employing various permutations of the terms and idioms: ‘arteriovenous malformation’, ‘AVM’, and ‘BAVM’, or ‘cavernous malformation’, ‘cavernoma’, and ‘cavernous angioma’ on the one hand, and ‘microRNA’, ‘miRNA’, and ‘miR’ on the other. Afterward, using cross-reference search, we then evaluated further articles pertaining to the individual miRNAs found in other cerebral diseases. We have considered both clinical studies on adult human patients and their pathological specimens, regardless of design, as well as experimental studies performed in vitro and in vivo. This non-systematic review may be significant in light of the increasing interest in the effective management and understating of cerebral vascular malformations.

## 3. microRNAs Involved in Cerebral Vasculogenesis and Angiogenesis

In order to better understand the pathophysiological implications of microRNAs in BAVMs and CCMs, we believe it necessary to first illustrate the mechanisms of normal vasculogenesis and angiogenesis within the brain. In the early embryonic life, the hemangioblasts emerge from the mesoderm as a response to local fibroblast growth factor (FGF) signaling. The surrounding endoderm secrets isoforms of vascular endothelial growth factor (VEGF) and the hemangioblasts express VEGFR2 which is essential for their survival and further differentiation. Once hemangioblasts differentiate, they will turn either into hematopoietic cells that will form the blood-islands or into angioblasts that will undergo vasculogenesis. The angioblasts are immature endothelial cells organized in clusters and express both VEGFR1 and VEGFR2. While VEGFR2 is responsible for proliferation, differentiation, and survival, VEGFR1 has opposite effects and supports the formation of the vascular lumen [23], shaping the primitive blood vessels (Figure 1). Research has been conducted on the effects of miRNAs on vasculogenesis by studying mutant variants with loss of function of the Dicer enzyme, an RNase involved in miRNA maturation. Notably, while knockout of VEGFR2 is correlated with the lack of primitive vascular plexi [24], in Dicer^ex1/2^ mice, even if the blood vessels are thin and disorganized, the presence of the vascular bedding suggests that miRNAs are not crucial for vasculogenesis, but for the next phase of vascular development: angiogenesis [25]. Indeed, Dicer^ex1/2^ mice associated lower levels of Tie1 while VEGF, VEGFR1 and VEGFR2 were upregulated, but this is explained by the formation of abnormal blood vessels that favor local hypoxia and respond via VEGF activation accordingly.

After vasculogenesis, the endothelial cells (ECs) from the primary vascular plexus generate new capillaries by sprouting and non-sprouting angiogenesis. Sprouting angiogenesis is typically seen in brain formation and it is also the prototype for wound healing and tumor angiogenesis whereas non-sprouting angiogenesis involves splitting and fusion of capillaries and mainly occurs in lung development. In physiological sprouting angiogenesis, FGF and VEGF stimulation of the ECs induces the activation of matrix metalloproteinases (MMPs): MMP-2 and MMP-9, thus promoting the lysis of the basement membrane and the extracellular matrix (ECM) in the vicinity [26]. VEGF binding to VEGFR also induces endothelial nitric oxide synthase (eNOS) phosphorylation which in turn generates nitric oxide. The nitric oxide will further nitrosylate β-catenin which dissociates from VE-cadherin. Normally, the VE-cadherin/β-catenin complex increases the expression of claudin-5 via PI3K/AKT pathway [27]. In this context, VEGF promotes the destabilization of the EC–EC junctions. Therefore, VEGFR2 expressing ECs proliferate and migrate towards the VEGF gradient maintained by the periventricular layer of the neural tube [28]. Moreover, VEGF stimulates the upregulation of integrins (α1β1, α2β1, and αvβ3) which facilitate EC–ECM interactions during the migration. Integrin α5β1 is induced by FGF, IL-8, and TNF-α and potentiate integrin αvβ3-mediated migration [29]. The leading ECs secrete platelet-derived growth factor (PDGF) which attracts pericytes and promotes the basement membrane assembly on the migration tract. Not only FGF and VEGF are important in sprouting angiogenesis, but also angiopoietins (Ang1, Ang2, and Ang4) which bind to Tie1 and Tie2 receptors to modulate the response to VEGF. Ang1/Tie2 interaction inhibits the vascular leakage and promotes EC sprouting which is pivotal considering Tie2 knockout mice do not develop brain capillaries and consecutively die in utero [30]. Conversely, in the presence of physiological Ang1 concentrations, Ang2 acts in an opposite manner and leads to vessel destabilization due to the fact that it cannot dissociate the inhibitory Tie1/Tie2 heterodimer, but in the absence of Ang1, Ang2 behaves as a weak agonist of Tie2 [31]. In response to VEGF, Ang2 is released from ECs and orchestrates their plasticity.

Subsequently to the generation of the vascular network, remodeling, pruning and maturation phenomena occur in order to form the functional adult circulatory system. In adult life, angiogenesis is restricted to wound healing, inflammation, and reproduction. In the absence of these triggers, ECs are maintained in a quiescent stage (Figure 1) by the balance between pro-angiogenic (VEGF, FGF, EGF, IFN, MMPs, ANGs) and anti-angiogenic (angioarrestin, angiostatin, IL-12, TSP-1, fibronectin fragment) factors. The network of these stimuli influences the local miRNAome and is also under the control of miRs. Therefore, in physiological conditions, the most abundant miRs in ECs are both pro-angiomiRs (miR-21, miR-126, let-7 family, miR-23, and miR-17-92 cluster) and anti-angiomiRs (miR-221, miR-222, and miR-24) [32].

miR-126 is considered an endothelial-specific miR and acts by downregulation of the putative genes SPRED1, PIK3R2, and VCAM-1. SPRED1 is an inhibitor of MAPK/ERK pathway, which is activated by upstream signals mediated by VEGFR2. VEGFR2 also activates the PI3K/AKT signaling pathway whereas PIK3R2 inhibits PI3K activation. Throughout these mechanisms, miR-126 facilitates the VEGFR2-mediated migration of ECs. The let-7 family also influences the migration by modulation of VEGF signals. For instance, let-7 family members possibly target thrombospondin-1 (TSP-1), an endogenous inhibitor of angiogenesis. TSP-1 blocks the VEGF pathway by several mechanisms including inhibition of VEGFR2 phosphorylation, direct binding to VEGF, and binding to the inactive form of MMP-9 [33]. Dicer and Drosha knockout in human umbilical vein endothelial cells (HUVECs) results in the upregulation of TSP-1, probably by the decrease of let-7 family [34]. Another study revealed that Dicer blockade influences angiogenesis through miR-17-5p and let-7b which reduce tissue inhibitor of metalloproteinase 1 (TIMP1) expression and consequently MMPs activity [35]. The other two members of the miR-17-92 cluster, namely miR-18a and miR-19a, were also demonstrated to repress proteins containing thrombospondin type 1 repeats (TSR), angiogenesis inhibitors related to TSP-1 [36]. miR-21, a well-known onco-miR, has angiogenic effects by targeting PTEN, an upstream regulator of both MAPK/ERK and PI3K/AKT pathways, and also by enhancing the expression of HIF-1α and VEGF [37]. Another direct target of miR-21 is TIMP3, whose downregulation promotes MMP-2 and MMP-9 secretion [38]. Regarding anti-angiomiRs, miR-221 and miR-222 have relatively similar targets including c-kit, Ets1, Ets2, ZEB2, STAT5a, and eNOS [39]. ECs express both c-kit and its ligand stem cell factor (SCF) and c-kit activates PI3K/AKT, MAPK/ERK, and JAK/STAT pathways. miR-24 inhibits eNOS, GATA2, PAK4, Delta-like ligand 1 (Dll1), and Notch1 which regulates PDGF, VEGF, and the corresponding receptors, determining proliferation and pericytes recruitment [40].

In hypoxic conditions, HIF-1α contributes to the alteration of miRNAs in HUVECs by upregulating miR-210, let-7a, let-7e, miR-103, miR-107, and to a lesser extent miR-150 and miR-328 [41,42]. miR-210 has been shown to directly target Ephrin A3 (EFNA3), modulating the activity of PI3K/AKT pathway. However, miR-328 is an anti-angiomiR whose target CD44 is involved in EC-EC and EC-ECM interactions [43]. Strikingly, let-7a, let-7e, miR-103, and miR-107 have AGO1 as a putative target and by the inhibition of AGO1-RISC complex, they produce a release in mRNAs that are suppressed during normoxia. Consequently, these miRs facilitate the VEGF desuppression in hypoxia (20). Under VEGF stimulation, miR-155, miR-191, miR-21, and some of the miR-17-92 cluster (miR-18a, miR-17-5p, and miR-20a) are overexpressed in ECs, also shifting the balance towards angiogenic factors [44].

## 4. microRNAs in Brain Arteriovenous Malformations

In the following paragraphs, we review the miRNAs associated with BAVMs, and also discuss the other pathologies these strands have been linked to. Because the involvement of both miR-137 and miR-195* in BAVMs was researched in the same study [20] and because they possess comparable effects in similar pathologies, they are covered within the same subsection.

### 4.1. miR-7-5p

Despite the fact that little is known of the implications miR-7-5p has in the development of BAVMs, Chen et al. established that it was significantly upregulated in the serum of patients with these malformations [18]. According to them, the Vascular Endothelial Growth Factor (VEGF) signaling pathway is implicated in 15 of the genes targeted by miR-7-5p, among miR-199a-5p and miR-200b-3p. Several studies showed that miR-7-5p is downregulated in the peripheral blood of glioblastoma patients [45], glioma cell lines and human glioma tissue [46,47], or in the microvasculature of these tumors themselves and targeting the RAF1 oncogene [48]. It has also been proven as protective against ischemia-reperfusion injury in in vivo rat models by some [49], and as aggravating by others [50]. A recent study demonstrated that miR-7-5p was decreased in the serum of patients suffering from hemorrhagic stroke [51]. Should the results extrapolated from rats be accurate, this would mean a diminished inhibition the PI3K/AKT pathway, leading to an increase of aquaporin 4 (AQP4) within the brain edema area caused by the hemorrhage. This was thought to aggravate to edema, while the rats receiving intraperitoneal bultylphthalide presented increased miR-7-5p expression and lower severity brain edema. Considering the evidence at hand, it is unclear whether miR-7-5p stimulates or represses angiogenesis within the developing brain or in a budding congenital vascular lesion. Its involvement in BAVMs other than a hypothetical peripheral biomarker remains to be ascertained.

### 4.2. miR-18a

Brain endothelial cells (BECs) of BAVMs are considered strikingly active, proliferate and migrate faster than normal endothelial cells, and exhibit atypical functions aside from modified expression levels of growth factors modulating angiogenesis [9,22,52]. Increased levels of thrombospondin-1 (TSP-1), an antagonist of Vascular Endothelial Growth Factor A (VEGF-A) have been discerned within BAVM BECs, as a result of the high levels of “inhibitor of DNA-binding protein A” (Id-1), which represses the transcription of TSP-1 [19,20,53]. With the aim of downregulating this inhibitor, the intensification via transfection of miR-18a, of the miR-17 to miR-92 cluster, might hamper the chaotic and unfettered growth of BAVMs, even following subtotal surgical resection or embolization. Although miR-17, miR-18a, miR-19a, and miR-20a have displayed antiangiogenic and tumor suppressive activity, most notably when considered independently, they have also been connected to oncogenesis and neoplastic angiogenesis [17,54,55]. The majority of members belonging to the miR-17-92 cluster were upregulated in pediatric brain tumors including medulloblastomas, ependymomas, and astrocytomas, with the highest expression being noticed for miR-18a and miR-18b [56]. Based on the study of Miao et al., miR-18a elevated the permeability of the glioma blood–tumor barrier through the runt-related transcription factor 1 (RUNX1) mediated reduction in tight junction related proteins ZO-1, occludin, and claudin-5 [57]. High expression of miR-18a has been observed in glioblastoma tissues and cell lines, with neogenin, a close relative of Deleted in Colorectal Cancer, as its target [58]. By binding with its ligands in healthy tissues, neogenin adjusts a number of physiological roles such as cellular migration, adhesion, and differentiation, tissular morphogenesis, axon guidance, angiogenesis, and even apoptosis, being regarded as a tumor suppressor in several neoplastic lesions [59,60,61,62,63,64]. Nevertheless, high levels of neogenin have been noticed in diffuse intrinsic pontine gliomas, affecting tumor invasion [65]. Its deficiency was also associated with the Moyamoya-like vasculopathy [66], and its depletion in mouse brains was correlated with abnormally permeable blood vessels with a reduced blood flow, disrupted vascular basement membranes, a declined number of pericytes, compromised BEC barrier, and increased BEC proliferation [67]. It is possible that miR-18a influences neogenin expression in BAVMs, however, to the best of our knowledge, this has yet to be investigated. As demonstrated by Marin-Ramos et al., miR-18a reduces VEGF production by downregulating plasminogen activator inhibitor (PAI-1) levels, the expression of bone morphogenetic protein 4 (BMP4), and diminishing hypoxia inducible factor 1α (HIF-1A) levels [68]. Technically, this can be summarized by a diminished activation of Smad1/5 and Notch signaling. The deterrent in HIF-1A expression via miR-18a transpires under the influence of normoxia, which is ordinarily present in BAVMs. Furthermore, miR-18a also lessened the invasiveness of BAVM BECs, associated with a decline in matrix metalloproteinase (MMP) 2 and 9, and ADAM metallopeptidase domain 10 (ADAM10) levels. As Ferreira et al. indicated, the most obvious effects of miR-18a emerged on cells subjected to different arterial shear flow conditions, its signaling significantly lessening the release of VEGF-A and VEGF-D from BECs of BAVMs [19]. This research was also the first to determine that the internalization of miR-18a can be attained even without a transfection reagent (naked microRNA) in untransformed BECs, and not sacrificing its functionality. Therefore, treatment with miR-18a might be successfully performed in patients with BAVMs in the near future, as long as its oncogenic properties do not lead to the development of cerebral tumors.

### 4.3. miR-137 and miR-195*

As stated by Huang et al., miR-137 and miR-195* within the vascular smooth muscle cells (VSMC) of BAVMs seemingly possess inferior expression rates than normal [20]. The functional role of these two miRNAs is the repression of cellular migration and tube formation, thereby reducing the survival of VSMCs in vitro. There is a likelihood that this mechanism deters anomalous angiogenesis and participates as a safeguard against the development of BAVMs. Nevertheless, inhibitors of miR-137 and miR-195* did not influence the aforementioned processes, inferring a low baseline level of these two miRNAs. Since the smooth muscles of BAVM vascular walls are hypertrophied, it seems likely that VSMCs are concerned, the actual cause of inflammation, angiogenesis and atheromatosis being the signaling interaction between VSMCs and various other cell types. miR-137 is reportedly involved in alleviating ischemic stroke via the lncRNA GAS5/miR-137 signaling pathway [69], by reducing JAK1 and STAT 1 production [70], or by regulating *Src*-related oncogenic signaling and disabling the mitogen-activated protein kinase (MAPK) cascade [71]. Its overexpression has also been associated with schizophrenia in several studies [72,73,74,75,76,77]. Its level is diminished in glioma cells, miR-137 having been shown to suppress cancer cell invasion, proliferation, and ability to metastasize by deterring the self-renewal process of neoplastic cells and stimulating their differentiation [78,79]. It is highly probable that either Rac1 or Cox-2, targets of miR-137, are accountable for these effects in gliomas [80,81], whereas CSE1L behaves similarly in oligodendroglial tumors and is directly inhibited by the miRNA in question [82]. Since it has been observed that the incidence of gliomas is decreased in schizophrenic individuals, miR-137 expression might be, at least in part, responsible [83]. Huang et al. also uncovered that 54 proteins normally inhibited by miR-137 and miR-195* are overexpressed in BAVM VSMCs, subject to specific signaling pathways that use protein kinase B (Akt), p38 mitogen-activated protein kinase (p38MAPK), phosphoinositide 3-kinase (PI3K), extracellular signal-regulated kinase (ERK), VEGF, and nuclear factor kappa-light-chain of activated B cells (NFkB), as mediators [20]. Regarding miR-195*, its downregulation leads to an increase of VEGFA in the infarcted brain [84], therefore its dwindled expression in BAVMs seems veritable. It has similarly exhibited a bolstered expression in schizophrenic patients, conducting toward a decreased brain-derived neurotrophic factor (BDNF) level and impaired cognitive functions, although this phenomenon has not been as widely studied [85,86]. On the other hand, Alzheimer’s disease (AD) patients with the lower cognitive performances demonstrated a diminished cerebral miR-195* presence, and elevating it rescued AD-related lysosomal irregularities in inducible pluripotent stem cells (iPSCs)-derived brain cells [87]. Furthermore, it has the capacity to lessen the beta-site APP cleaving enzyme 1 (BACE1) protein level and amyloid-β in mouse brains [88], and to limit amyloid-β accumulation, dendritic degeneration, and neuronal death in rats with chronic brain hypoperfusion [89,90], suggesting that it could be a veritable therapeutic agent for AD. miR-195* ameliorated neural apoptosis in rats subjected to ischemic stroke by suppressing the KLF5-mediated activation of the JNK signaling axis [91], while also displaying the potential to block inflammatory processes via NFkB and promote neural stem cell proliferation and migration after stroke [92]. Lastly, it has also shown tumor suppressive effects in malignant meningiomas by inhibiting fatty acid synthase (FASN) [93], as well as stimulating apoptosis and reducing proliferation and migration in gliomas [94,95]. Currently, it is unclear which of the aforementioned pathways and mechanisms affected by miR-137 and miR-195* are directly involved in BAVM pathophysiology.

### 4.4. miR-199a-5p

The miRNA in question has exhibited diverse behaviors within different systems and diseases [96]. Two loci of the human genome are tasked with its encoding, namely miR-199a-1 found in chromosome 19, and miR-199a-2 in chromosome 1 respectively, both of them producing the exact same miRNAs (miR-199a-5p and miR-199a-3p). The discrepant methylation of these two loci was observed to result in differential expression levels in gliomas and testicular germ cell tumors, with overexpression in the former and downregulation in the latter [96]. Aside from gliomas, an upregulation of miR-199a-5p has also been observed in cerebral metastases of colorectal cancer [97]. Nevertheless, in the research performed by Wang et al., it was illustrated that microRNA-199a-5p repressed glioma progression, invasion, and migration in vitro by directly targeting magnesium transporter 1 (MAGT1) [98], whereas according to Zhang et al. the same effect was obtained by targeting the membrane-associated ring-CH-type finger 8 (MARCH8) [99]. In glioblastomas, the upregulation of miR-199a-5p by circPVT1 silencing hindered viability, and migration, while also positively influencing apoptosis [100]. Considering the divergent functions of miR-199a-5p concerning brain tumors, it is possible that multiple pathways with opposing effects are also involved in BAVMs. Zhong et al. found increased miR-199a-5p expression in vitro an in vivo mouse models subjected to brain ischemia via MCA occlusion, as opposed to the long non-coding RNA (lncRNA) ANRIL, which was downregulated [101]. They established that either ANRIL overexpression or miR-199a-5p repression could offer a protective effect against cells against cellular ischemic injury via the CAV-1 mediated MEK/ERK pathway, which is directly targeted by miR-199a-5p. Contrariwise, the results by Li et al. suggested that miR-199a-5p itself was protective against cerebral ischemic injury in rats reducing discoidin domain receptor 1 (DDR1) expression [102]. Another study showed that the upregulation of this microRNA helped protect against the ischemia/reperfusion-induced injury in the spinal cord of rats by silencing the ECE1 gene, subsequently constraining caspase-9, Bcl-2, p-JNK, and p-ERK production [103]. As such, the exact role this microRNA plays in triggering or protecting neural cells from ischemia is uncertain. Additionally, miR-199a-5p is known to suppress cellular proliferation and promote apoptosis in hemangiomas by way of HIF-1A signaling [21]. It is also apparent that its inhibition via the lcnRNA H19 escalates the angiogenic potency of mesenchymal stem cells (MSCs) concomitantly while increasing VEGF-A expression [104]. As reported by Chen et al., miR-199a-5p was consistently upregulated in the serum of BAVM patients, acting via the VEGF pathway [18]. This miRNA can also be utilized as a biomarker for brain AVMs, along with miR-7-5p and miR-200b-3p. In agreement with the findings of Heuslein et al., miR-199a-5p acted as a strong regulator of arteriogenesis, as well as a putative target in the management of peripheral arterial disease [105]. In this study, it suppressed the expression of pro-arteriogenic genes and the in vitro adhesion of monocytes onto endothelial cells, whereas in vivo its overexpression affected both perfusion and arteriogenesis in the limbs of mice. It remains to be demonstrated whether miR-199a-5p has a singular and consistent role in BAVM pathophysiology, or whether it has multiple and possibly conflicting implications.

### 4.5. miR-200b-3p

An upregulation of miR-200a-3p noticed in the peripheral blood could explain certain features of BAVM pathophysiology, especially in individuals with degenerative alterations of the intrinsic vessels [18]. Zhang et al. showed that miR-200b-3p was also upregulated in neonatal rat brains in the early stage following hypoxia-ischemic injury, and that its inhibition led to a diminished neuronal apoptotic rate [106]. Medulloblastomas exhibit differential expression of miR-200b-3p according to age, with adults presenting an overexpression compared to pediatric patients [107]. Furthermore, miR-200b-3p among other miRNAs was upregulated in patients with brain metastatic gastric adenocarcinomas when compared to patients with this cancer but without cerebral metastases [108]. ZEB2, also referred to as Smad-interacting protein 1 (SIP1) and a known suppressor of the cell-to-cell adhesion molecule E-cadherin [109], was established as the primary target of miR-200b-3p, presenting significant downregulation. According to Li et al., the same mechanism is available for gliomas [110]. Glioma tissues and human glioma cell lines were shown to also have a diminished expression of miR-200b-3p, whereas its increase led to E-cadherin upsurge, declined mesenchymal markers, and a dwindled neoplastic cellular proliferation, migration, and invasion in vitro [111]. The target of this miRNA was identified as extracellular-regulated protein kinase 5 (ERK5). In vivo, miR-200b-3p also demonstrated its suppressive influence on tumor expansion. Hu et al. revealed that miR-200b also regulates the production of lactate dehydrogenase A in gliomas [112], while a recent metanalysis concluded that miRNA-200 expression level might be negatively correlated with the WHO glioma grade of gliomas [113]. The difference of miR-200b-3p expression between gliomas and BAVMs might insinuate that the neoplastic vessels arising in former arise via a different signaling cascade than the ones that constitute the latter. Mouse brains with folate deficiency presented a reduction in miR-200b-3p, alongside miR-106a-5p and miR-339-5p, eliciting a stimulation of the Amyloid Precursor Protein (APP) gene and the subsequent accumulation of amyloid-β [114]. It was suggested that transfection with miR200b/c might inhibit amyloid-β secretion and consecutive cognitive impairment by improving insulin signaling [115]. This could potentially explain one of the pathophysiological mechanisms of AD. Analogously, the brains of mice infected with prions had a lessening in the majority of microRNAs belonging to the miR-200 cluster, including miR-200b-3p, as well as an increase of several of the miRNAs deregulated in AD [116]. The neuroprotector thymosin beta 4 showed promise in mouse models subjected to severe traumatic brain injury, raising the serum levels of both miR-200a-3p and miR-200b-3p [117]. Interestingly, miR-200b-3p and miR-200c-3p have been identified as indicators of cytopathic inflammation produced by cytomegalovirus (CMV) infection [118]. Nonetheless, the means through which this miRNA contributes to the pathophysiology of BAVMs remain ambiguous.

### 4.6. let-7b-3p

As indicated by Chen et al., increased serum levels of let-7b-3p have been observed in BAVM patients’ serum, although its mechanism is as of yet unclear [18]. Contrarywise, it was significantly downregulated in the serum of individuals with CCMs, as determined in a recent genome-wide association study (GWAS) [119]. In this study, it was determined that let-7b-3p targeted 981 genes, out of which endoglin (ENG), Hes family bHLH transcription factor 1 (HES1), HIF-1A, MAPK1, mindbomb E3 ubiquitin protein ligase 1 (MIB1), occludin (OCLN), programmed cell death 10 (PDCD10), and tight junction protein 1 (TJP1) were relevant to CCM formation. As such, it may be utilized in the future as a prospective diagnostic marker, or even as an angiogenic inhibitor, should its function be fully elucidated. let-7b-3p is particularly downregulated in seizures, along with several other miRNAs [120,121], and is hypothesized to act as a key regulator of various pathways correlated to mesial temporal lobe epilepsy [122], possibly being also implicated in the epileptic manifestation of BAVMs. This assumption demands validation in an upcoming research. After induced middle cerebral artery ischemic stroke in rats, Li et al. ascertained that let-7b-3p had a low expression in both blood and penumbra brain tissue, whereas another microRNA, namely miR-223-3p, had an increased level both in serum and ischemic penumbra [123]. Moreover, let-7b-3p is believed to possess a tumor repressing ability in gliomas, its decreased expression being correlated to a poorer prognosis in patients with these lesions via a disturbed suppression of inhibitor of nuclear factor kappa-B kinase subunit epsilon (IKBKE), E2F2, and the oncogenes KRAS, HMGA2 and MYC [124,125,126]. Findings such as these may highlight that BAVMs and CCMs have distinct pathological cascades that potentially converge at a crossroad, the only difference being the over- or underexpression of a certain microRNA such as let-7b-3p. Finally, the amplified amount of let-7b-3p in patients BAVMs could be the result of the increased blood flow within the affected areas of the brain, based on the observation that its serum levels were diminished after infarct [123]. As CCMs do not possess such a high intralesional flow as BAVMs, and also incorporate areas of reactive gliosis and microhemorrhages, a local response mimicking cerebral ischemia may be the reason this miRNA is downregulated in cavernoma patients. A synopsis of the miRNAs presented, can be found in Table 1, as well as their relative expression levels. Figure 2 shows the currently known pathways of BAVM development wherein miRNAs are involved.

## 5. microRNAs in Cerebral Cavernous Malformations

Below, the miRNAs linked to CCMs are discussed, along with other illnesses possibly influenced by them. miR-27a, miR-125a, and mmu-miR-3472a have been individually studied in several other diseases, as well as in association with other similar acting microRNAs. The genome-wide association study performed by Kar et al. disclosed 52 dysregulated miRNAs in brainstem CCM patients compared to controls, 10 of which were upregulated and 42 downregulated [119]. A more stringent statistical analysis narrowed down five miRNAs (let-7b-5p, miR-361-5p, miR-370-3p, miR-181a-2-3p, and miR-95-3p) as the most significant for CCM, which are discussed ulteriorly in this section, apart the previously reviewed let-7b-5p.

### 5.1. miR-27a

In the study by Li et al., miR-27a was noticed to be increased in BECs isolated from in vivo mouse model brains harboring early CCM lesions and in cultured BECs displaying either CCM1 or CCM2 depletion [127]. miR-27a acts as an inhibitor of vascular endothelial (VE)-cadherin, and therefore presumably results in a severe disruption of BEC junctional integrity bordering the cavernomas. Additionally, miR-27a operates downstream of Krüppel-like factor (KLF) 2 and 4, whose exaggerated expression are recognized as causative of CCM development independent of VE-cadherin [119,127,128,129]. Knockdown of either KLF2 or KLF4 via small interfering RNA (siRNA) partially decreased miR-27a up-regulation within cultured BECs depleted of CCM2, whereas the simultaneous inhibition of KLF2 and KLF4 completely negated the up-regulation of miR-27a in these cells [127]. This suggests the existence of a regulatory pathway through which the loss of one of the three CCM genes stimulates KLF2/4, resulting in increased miR-27a, which in turn might diminish VE-cadherin expression. miR-27a-3p downregulation in rats was also linked to the upsurge of mitogen-activated protein kinase 4 (MAP2K4) production, resulting in a suppression of hippocampal neuronal apoptosis in epilepsy, both in vivo and in vitro [130]. On the other hand, the intraventricular delivery of a miR-27a-3p mimic in rats suffering from collagenase-induced intracerebral hemorrhage induced a mitigation of brain edema, microvascular leakage, and leukocyte infiltration, as well as an inhibition of neuronal apoptosis and microglia activation within the perihematomal zone by suppressing the overexpression of aquaporin-11 in BECs [131]. Similarly, the induced overexpression of miR-27a in rat brains before traumatic brain injury markedly diminished the neurological deficits and cerebral injury, particularly repressing autophagic activation after trauma occurred [132]. Based on the findings of Sun et al., this happened due to the direct targeting of the Forkhead box O3a (FoxO3a) protein expression and its subsequent suppression. A comparable research performed on mice showed that the administration of mimics of miR-27a along with miR-23a considerably reduced the activation of proapoptotic Bcl-2 family constituents Bax, Noxa, and Puma, lessened the severity of cortical lesion volume and hippocampal neural cell loss, and also undermined markers of caspase-dependent and -independent apoptosis after traumatic brain injury [133]. In the study by Raoof et al., one of the three miRNAs uncovered with biomarker potential in temporal lobe epilepsy in adult patients was miR-27a-3p, the other two being miR-328-3p and miR-654-3p [134]. It may be possible that glycine receptor alpha 2 (GLRA2), with a heightened intracerebral expression and linked to epilepsy is the pathway through which miR-27a-3p operates, an aspect that may also be extrapolated to CCM-induced seizures. Transfection of human BEC lines with miR-27a-3p and miR-222-3p led to a decrease in phosphodiesterase 3 (PDE3) expression, representing a potential link to signaling pathways pertinent to cerebrovascular integrity and healing capacity [135]. According to Tripathi et al., the increased levels of mir-27a was associated with the blockage of oligodendrocyte precursor cell proliferation and differentiation, as well as deficient myelination and remyelination via Wnt-β-catenin signaling disruption [136]. This evidence might suggest a dual role of miR-27a-3p, being both neuroprotective and neurodestructive, according to the targeted pathway. More studies need to be performed on patients harboring CCMs in order to validate these proposed mechanisms and clinical implications.

### 5.2. miR-125a

miR-125a is essential to the formation of the BEC barrier and is decreased in the affected brains of mice with the deletion of either CCM1 or CCM2 genes [127]. The downregulation of miR-125a has also been observed in children with epilepsy, despite the fact that no connection could be established between this miRNA and the inflammatory cytokines IFN-γ and TNF-α [137]. In rats, miR-125a was decreased in the blood of all models presenting seizures triggered by ischemia, hemorrhage, or kainite toxicity [138], as well as within the hippocampus after pentylenetetrazol-induced epilepsy [139]. In humans, mir-125a-5p has already been identified as a potential biomarker for ischemic stroke [140]. Furthermore, the overexpression of miR-125a-5p resulted in a reduction of seizures and lessened hippocampal inflammation via the suppression of its target gene, calmodulin-dependent protein kinase IV (CAMK4) [139]. Pan et al. showed that microvesicles carrying miR-125a-5p fostered the proliferation, migration, the ability to form tubes of transfected microvessel BECs, improved their apoptotic rate [141], while an earlier study discovered the role of this microRNA in protecting BECs from the injurious effects of oxidized low-density lipoproteins such as, senescence, decrease of nitric oxide (NO), and upsurge of reactive oxygen species (ROS) [142]. It is therefore likely that this microRNA plays a distinctive role in CCM patients presenting with epilepsy, or it could hypothetically act as a biomarker in identifying the cases liable to develop CCM-stimulated seizures.

### 5.3. mmu-miR-3472a

Koskimäki et al. determined that mmu-miR-3472a presented a differential expression in the serum of mice with or without the loss of the Ccm3/Pdcd10 gene [143]. More interestingly, they discovered that mmu-miR-3472a has cullin associated and neddylation dissociated 2 (CAND2) as the putative target, which was often dysregulated in the transcriptomes of all three models researched (acute, chronic, and control). Also in this study, CAND2, which encodes TIP120b, a muscle-specific TATA-binding protein that is crucial in the process of myogenesis [143,144,145], was downregulated within in vitro BECs, though upregulated within the in vivo neurovascular units (NVUs) of acute and chronic models [143]. CAND2 is associated with the signaling pathway of cullins [146], also coordinating the activity of the Skp, Cullin, and F-box containing complex (SCF complex). Moreover, cullins are known to control the levels of VEGF2R on the surface of vascular endothelial cells via mRNA expression [147], as well as modulate the function of the endothelial barrier through the expression of vascular endothelial (VE)-cadherin, a key constituent of endothelial cell-to-cell adherens junctions. [148]. These factors imply a potential role of CAND2 and mmu-miR-3472a in the development of CCMs. Nevertheless, further research needs to be conducted in order to establish a pathophysiological relationship.

### 5.4. miR-361-5p

This microRNA was observed as downregulated in the brainstem cavernomas included in the recent GWAS, with possible functional significance [119]. miR-361-5p has been established as a VEGF modulator in coronary artery disease [149], is implicated in the functionality of the blood–brain barrier including the formation of adherens junctions, lymphocyte adhesion to endothelial cells, and the integrity of the extracellular matrix [150], as well as an overexpressed serum biomarker in patients with acute myocardial infarction, ischemic stroke, and pulmonary embolism [151]. In patients with multiple sclerosis, its overexpression has also been correlated with decreased disease severity, suggesting a neuroprotective effect [150]. Among other miRNAs, miR-361-5p behaved as a crucial modulator of X-linked and autosomal genes of intellectual disability [152]. Because of these multiple involvements, miR-361-5p may represent a valid therapeutic target or biomarker for CCMs, although more research on its exact mechanisms is needed.

### 5.5. miR-370-3p

Regarding brainstem CCMs, miR-370-3p might also present biological or pathophysiological importance for patients harboring lesions at this level by targeting the gene Neurofibromatosis type 1 [119]. Lulli et al. have shown that miR-370-3p expression is markedly downregulated in gliomas when compared to normal brain tissue and neural stem cells [153]. The recovery of miR-370-3p expression in glioblastoma stem-like cells considerably reduced their proliferation, migration, and clonogenic abilities both in vitro, and in vivo models. Amid the genes negatively influenced by the normalized expression of miR-370-3p, EMT-inducer high-mobility group AT-hook 2 (HMGA2), HIF-1A, and the lncRNAs Nuclear Enriched Abundant Transcript (NEAT)1 were ascertained. Additionally, miR-370-3p directly targets the 3′ untranslated region of β-catenin mRNA, consequently inhibiting the expression of the canonical Wnt cascade downstream oncogenes cyclin D1 and c-myc in human gliomas, as well as induces cell cycle arrest within these malignant cells [154]. Moreover, the miR-370-3p-Wnt7a axis modulates epithelial-mesenchymal transition in glioma cells in vitro as well as in vivo, and knockdown of the lncRNA CTBP1-AS2 hindered this process along with proliferation and migration [155]. This translates into a marked repressive effect on glioma cell proliferation by miR-370-3p overexpression or rescue. In an in vivo study, high levels of miR-370-3p were obtained in both brains and serum of mice with sepsis-associated encephalopathy (SAE), though not in sepsis alone or uremia, advocating a specificity towards this type of cerebral lesion [156]. It was hypothesized that TNF-α was responsible for vulnerating neurons to apoptosis in SAE at least partly through miR-370-3p induction. Another possible implication of miR-370-3p could be in primary progressive multiple sclerosis, wherein patients demonstrated significantly dysregulated serum levels of several microRNAs compared to controls, including this one [157]. Whether miR-370-3p downregulation also impacts the formation of CCMs by interfering with inflammatory processes within the brain remains to be validated.

### 5.6. miR-181a-2-3p

Aside from the downregulation in CCM patients [119], low levels of miR-181a-2-3p, which targets FOXP2, have been linked to a poorer prognosis in patients with glioblastoma, although the exact acting mechanism of FOXP2 in these malignant tumors remains unclear [158]. miR-181a-2-3p is also downregulated under hypoxic conditions in glioblastoma cell lines in vitro [159] and has displayed a statistically significant differential expression between grade III and grade IV glioma cells, with an inferior level within the latter [160]. However, we could not find any additional information correlating this miRNA to other cerebrovascular malformations.

### 5.7. miR-95-3p

Little is known of the implications miR-95-3p has in pathologies of the brain outside its discovered downregulation in CCM patients recruited in the GWAS by Kar and associates [119]. In an earlier research, it had been shown that miR-95-3p levels were positively correlated to glioma grade, and that a reduction in this microRNA elicited a similar response regarding the ability of glioma cells to proliferate and invade, but enhanced apoptosis [161]. An RNA-binding protein, CUGBP- and ETR-3-like family 2 (CELF2), was incriminated as the presumable target of miR-95-3p, being a likely tumor suppressor. Conversely, the overexpression of miR-95-3p caused a decrease in invasiveness, proliferation, and clonogenicity of metastatic lung adenocarcinoma within the brain by inhibiting oncogene cyclin D1 expression, and as a result prolonging metastasis-free survival [162]. Future studies may shed a light on whether miR-95-3p possesses a pathophysiological role in CCM development, or if its downregulation in this type of lesion was merely incidental. Table 2 summarizes the miRNAs discovered in CCMs, time of discovery, relative expression levels, and possible implications, and Figure 3 illustrates the current findings regarding their signaling pathways and putative targets.

## 6. Limitations

Currently, there are significant gaps in our understanding of the roles miRNAs play in the development of intracranial vascular malformations, and the associations described in our review may at times be vague or conjectural. This major limitation is most likely the result of the rarity of these lesions, as well as the definitive and curative nature of the principal treatment methods available, especially surgical removal. As such, a more exhaustive research into the molecular, genetic, and epigenetic mechanisms implicated in their pathophysiology might not seem as attractive as those of incurable and malignant cerebral diseases, where findings in this field may become groundbreaking game changers. However, we are hopeful that the information provided herein would eventually serve as a steppingstone or a solid basis for upcoming research that may provide associations between BAVMs or CCMs and various other brain pathologies. There is a likelihood that by unfolding the roles of miRNAs and their targets will help us in better understanding these malformations and their process of evolution. Should the functions of miRNAs be elucidated, it may also be possible to soon prevent or hinder the development of BAVMs or CCMs, and thus avoid possible life-threatening complications such as severe intracranial hemorrhage, or at least alleviate the complexity of the neurosurgical procedure.

## 7. Conclusions

Both BAVMs and CCMs are the product of multifarious molecular and genetic factors, with hemodynamic, structural, and degenerative alterations supplying a crucial contribution. However, because of their rarity, studies on the genetic and epigenetic factors involved in the occurrence of these lesions may not seem as appealing or critical as other more frequent cerebral pathologies. As such, our grasp on the miRNAs and affected signaling pathways involved is as of yet lacking, and sometimes based on conjecture. This review aimed to be a succinct and up-to-date summary on our current understanding of the roles of miRNAs and their potential as therapeutic targets or diagnostic biomarkers for BAVMs and CCMs. Despite the fact that many of the mechanisms described in other cerebral pathologies may not be actively involved in the development of intracranial vascular malformations, there is a possibility that at least some common links are present, which can provide valid treatment options in the future. Lastly, while surgery remains the definitive curative therapy for both BAVMs and CCMs, miRNA-targeted therapy may provide an adjuvant management strategy that could assist in neurological recovery after stroke, or hinder vascular malformation growth, thereby lessening the difficulty of the neurosurgical act.

## Figures and Tables

**Figure 1 cells-10-01373-f001:**
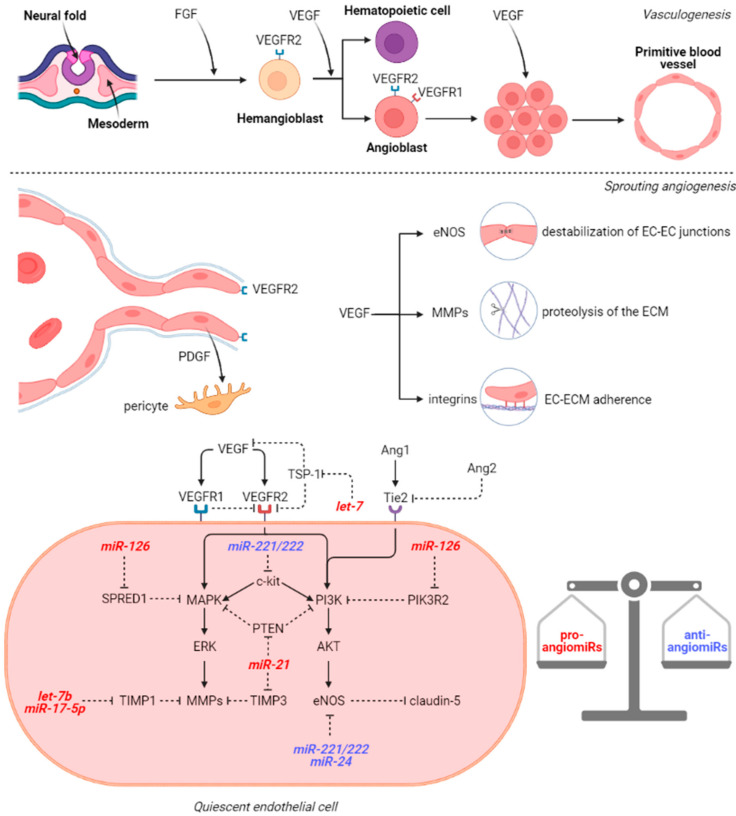
VEGF signaling pathway in vasculogenesis and angiogenesis. Abbreviations (in alphabetical order): AKT, Protein kinase B; EC, endothelial cell; ECM, endothelial cell membrane; eNOS, nitric oxide synthase; ERK, extracellular signal-regulated kinases; FGF, fibroblast growth factor; MMPs, metalloproteinases; MAPK, mitogen-activated protein kinase; miR, microRNA; PDGF, platelet-derived growth factor; PI3K, Phosphoinositide 3-kinases; PIK3R2, phosphoinositide-3-Kinase Regulatory Subunit 2; PTEN, Phosphatase and tensin homolog; SPRED1, Sprouty Related EVH1 Domain Containing 1; TIMP1/3, tissue inhibitor of metalloproteinases 1/3; VEGF, vascular endothelial growth factor; VEGFR, vascular endothelial growth factor receptor.

**Figure 2 cells-10-01373-f002:**
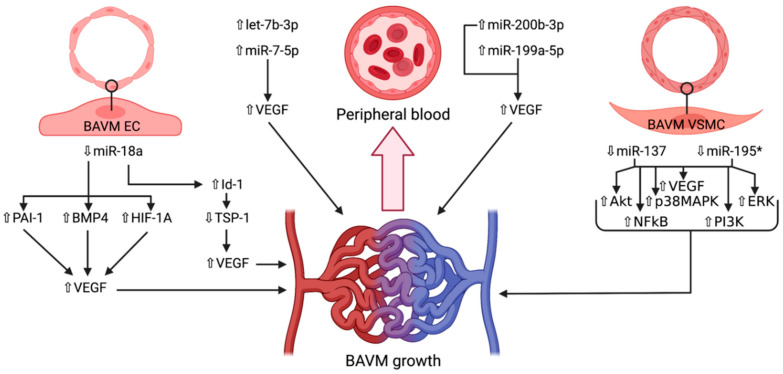
Schematic representation of the microRNAs and their respective expression levels, putative targets and pathways involved in the development of brain arteriovenous malformations. The microRNAs are divided according to the location wherein they were identified: brain endothelial cells, vascular smooth muscle cells, or peripheral blood. An upward pointing transparent arrow signifies upregulation; a downward pointing transparent arrow denotes downregulation. Abbreviations (in alphabetical order): *Akt*, protein kinase B; *BAVM*, brain arteriovenous malformation; *BMP4*, bone morphogenetic protein; *EC*, endothelial cell; *ERK*, extracellular signal-regulated kinase; *HIF-1A*; hypoxia inducible factor 1α; *Id-1*, inhibitor of DNA-binding protein A; *miR*, microRNA; *NFkB*, nuclear factor kappa-light-chain of activated B cells; *p38MAPK*, p38 mitogen-activated protein kinase; *PAI-1*, plasminogen activator inhibitor 1; *PI3K*, phosphoinositide 3-kinase; *TSP-1*, thrombospondin 1; *VEGF*, vascular endothelial growth factor; *VSMC*, vascular smooth muscle cell.

**Figure 3 cells-10-01373-f003:**
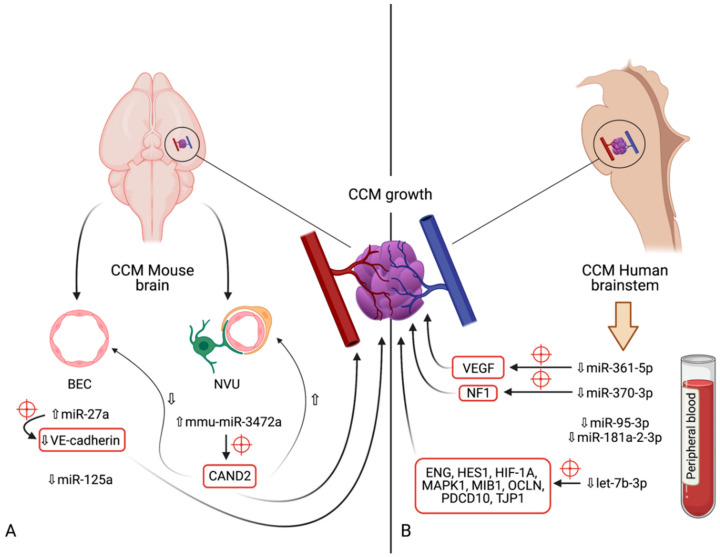
Diagram indicating the microRNAs and their respective expression levels, putative targets (marked by red crosshairs) and pathways involved in the development of cerebral cavernous malformations. The microRNAs are divided according to the model in which their expression was determined: (**A**)—mouse models (in vivo); (**B**)—human brainstem (according to the genome-wide association study by Kar et al. [97]). An upward pointing transparent arrow signifies upregulation; a downward pointing transparent arrow denotes downregulation. Abbreviations (in alphabetical order): *BEC*, brain endothelial cell; *CAND2*, cullin associated and neddylation dissociated 2; *CCM*, cerebral cavernous malformation; *ENG*, endoglin; *HES1*, Hes family bHLH transcription factor 1; *HIF-1A*; hypoxia inducible factor 1α; *HMGA2*, high-mobility group AT-hook 2; *MAPK1*, mitogen-activated protein kinase 1; *MIB1*, mindbomb E3 ubiquitin protein ligase 1; *miR*, microRNA; *NVU*, neurovascular unit; *OCLN*, occludin; *PDCD10*, programmed cell death 10; *TJP1*, tight junction protein 1; *VE-cadherin*, vascular endothelial cadherin.

**Table 1 cells-10-01373-t001:** Location, Expression and Putative Targets of Identified microRNAs in Brain Arteriovenous Malformations.

Author	miRNA	Location	Expression in BAVM	Putative Targets (BAVM)	Known Cerebral Pathologies Associated (Expression)—Respective Putative Target(s)
Chen et al., 2018 [18]	miR-7-5p	Peripheral blood	Upregulated	VEGF	Glioma, glioblastoma (downregulated)—RAF Ischemia-reperfusion injury Hemorrhagic stroke (downregulated)—PI3K/AKT pathway
Ferreira et al., 2014 [19]	miR-18a	BAVM BEC	Downregulated	Id-1, TSP-1 (indirect)	Pediatric medulloblastoma, ependymoma, astrocytoma (upregulated)—RUNX1 Glioblastoma (upregulated)—neogenin Moyamoya-like vasculopathy (downregulated)
Marin-Ramos et al., 2020 [46]				PAI-1, BMP4, HIF-1A	
Huang et al., 2017 [20]	miR-137	VSMC	Downregulated	Akt, p38MAPK, PI3K, ERK, VEGF, NFkB	Ischemic stroke—lncRNA GAS5//JAK1 and STAT 1//Src and MAPK Schizophrenia (upregulated) Gliomas (downregulated)—Rac1 or Cox-2 Oligodendrogliomas (downregulated)—CSE1L
Huang et al., 2017 [20]	miR-195*	VSMC	Downregulated	Akt, p38MAPK, PI3K, ERK, VEGF, NFkB	Ischemic stroke—VEGFA schizophrenia (upregulated)—BDNF Alzheimer’s disease (downregulated)—BACE1//KLF5 signalling//NFkB Gliomas (downregulated) Malignant meningioma (downregulated)—FASN
Chen et al., 2018 [18]	miR-199a-5p	Peripheral blood	Upregulated	VEGF	Glioma (upregulated)—MAGT1//MARCH8 Testicular germ cell tumors (downregulated) Ischemic stroke (upregulated)—CAV-1 mediated MEK/ERK pathway//DDR1//ECE1 Hemangioma—HIF-1A
Chen et al., 2018 [18]	miR-200b-3p	Peripheral blood	Upregulated	VEGF	Ischemic stroke (upregulated) Medulloblastoma (upregulated) Gastric adenocrcinoma metastasis (upregulated)—ZEB2 Glioma (downregulated)—ERK5 Alzheimer’s disease mouse models (downregulated)—APP Cytomegalus infection (upregulated)
Chen et al., 2018 [18]	let-7b-5p	Peripheral blood	Upregulated	NS	CCM (downregulated) Epilepsy (downregulated) Ischemic stroke models (downregulated) Glioma (downregulated)—IKBKE, E2F2, oncogenes KRAS, HMGA2, and MYC

Abbreviations (in alphabetical order): *Akt*, protein kinase B; *APP*, amyloid precursor protein; *BACE1*, beta-site APP cleaving enzyme 1; *BAVM*, brain arteriovenous malformation; *BDNF*, brain-derived neurotrophic factor; *BEC*, brain endothelial cell; *BMP4*, bone morphogenetic protein 4; *CCM*, cerebral cavernous malformation; *DDR1*, discoidin domain receptor 1; *ERK*, extracellular signal-regulated kinase; *ERK5*, extracellular signal-regulated kinase 5; *FASN*, fatty acid synthase; *HIF-1A*; hypoxia inducible factor 1α; *HMGA2*, high-mobility group AT-hook 2; *Id-1*, inhibitor of DNA-binding protein A; *IKBKE*, inhibitor of nuclear factor kappa-B kinase subunit epsilon; *KLF5*, kruppel-like factor 5; *MAGT1*, magnesium transporter 1; *MAPK*, mitogen-activated protein kinase; *MARCH8*, membrane-associated ring-CH-type finger 8; *miRNA* and *miR*, microRNA; *NFkB*, nuclear factor kappa-light-chain of activated B cells; *NS*, not specified; *p38MAPK*, p38 mitogen-activated protein kinase; *PAI-1*, plasminogen activator inhibitor 1; *PI3K*, phosphoinositide 3-kinase; *RUNX1*, runt-related transcription factor 1; *TSP-1*, thrombospondin 1; *VEGF(A)*, vascular endothelial growth factor (A); *VSMC*, vascular smooth muscle cell.

**Table 2 cells-10-01373-t002:** Location, Expression and Putative Targets of Identified microRNAs in Cerebral Cavernous Malformations.

Author	miRNA	Location	Expression in CCM	Putative Targets	Known Cerebral Pathologies Associated (Expression)—Respective Putative Target(s)
Li et al., 2020 [105]	miR-27a	BEC (mouse model)	Upregulated	VE-cadherin	Temporal lobe epilepsy (upregulated)—GLRA2 Neuroprotection in rat models after TBI—Bax, Noxa, and Puma//FoxO3a
Li et al., 2020 [105]	miR-125a		Downregulated	NS	Epilepsy—CAMK4 Ischemic stroke
Koskimäki et al., 2019 [121]	mmu-miR-3472a	Mouse NVU	Dysregulated	CAND2	NS
Kar et al., 2017 [97]	miR-361-5p	Brainstem CCM	Downregulated	VEGF, EGFL7	Acute myocardial infarction Ischemic stroke Pulmonary embolism Multiple sclerosis Intellectual disability
Kar et al., 2017 [97]	miR-370-3p		Downregulated	NF1	Glioma (downregulated)—HMGA2, HIF-1A, NEAT1//CTBP1-AS2 Sepsis associated encepahlopathy in mouse models
Kar et al., 2017 [97]	miR-181a-2-3p		Downregulated	NS	Glioblastoma (downregulated)—FOXP2
Kar et al., 2017 [97]	miR-95-3p		Downregulated	NS	Glioma (upregulated)—CELF2 Lung cell adenocarcinoma metastasis (downregulated)—cyclin D1
Kar et al., 2017 [97]	let-7b-5p		Downregulated	ENG, HES1, HIF-1A, MAPK1, MIB1, OCLN, PDCD10, TJP1	BAVM (upregulated) Epilepsy (downregulated) Ischemic stroke models (downregulated) Glioma (downregulated)—IKBKE, E2F2, oncogenes KRAS, HMGA2 and MYC

Abbreviations (in alphabetical order): *BAVM*, brain arteriovenous malformation; *BEC*, brain endothelial cell; *CAMK4*, calmodulin-dependent protein kinase IV; *CAND2*, cullin associated and neddylation dissociated 2; *CCM*, cerebral cavernous malformation; *CELF2*, CUGBP- and ETR-3-like family 2; *EGFL7*, epidermal growth factor like domain 7; *ENG*, endoglin; *FoxO3a*, Forkhead box O3a; *GLRA2*, glycine receptor alpha 2; *HES1*, Hes family bHLH transcription factor 1; *HIF-1A*; hypoxia inducible factor 1α; *HMGA2*, high-mobility group AT-hook 2; *KLF2*, kruppel-like factor 2; *MAPK1*, mitogen-activated protein kinase 1; *MIB1*, mindbomb E3 ubiquitin protein ligase 1; *miRNA* and *miR*, microRNA; *NEAT1*, Nuclear Enriched Abundant Transcript 1; *NS*, not specified; *NVU*, neurovascular unit; *OCLN*, occludin; *PDCD10*, programmed cell death 10; *TBI*, traumatic brain injury; *TJP1*, tight junction protein 1; *VE-cadherin*, vascular endothelial cadherin.

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
