# Peer review of "An Insight into the microRNAs Associated with Arteriovenous and Cavernous Malformations of the Brain"

_cells, 2021, doi:10.3390/cells10061373_

Round 1

Reviewer 1 Report

The authors searched some databases about the link of some microRNAs and Arterio-venous or Cavernous Malformations and summarized their findings in this manuscript. I am sure the authors put a lot of time and efforts in literature search and manuscript preparation. Even if most of the evidence are vague and the association between the microRNA expression and the malformation development is quite weak. The information is still useful for researchers in this field. One suggestion is that the authors should concentrate on the Arterio-venous and Cavernous Malformations. The discussion about the link of the microRNAs with other conditions should be limited to the extent that is relevant to the main diseases.

Author Response

Cover letter for: cells-1233098

An Insight into the microRNAs Associated with Arterio-venous and Cavernous Malformations of the Brain

Esteemed editor and reviewers,

We, the authors, would like to express our gratitude for your kind remarks and suggestions for our manuscript. We hope that the changes made are to your expectations. If, however, you would like further changes, we are more than happy to comply. As instructed by the editor, we have also provided a Graphical abstract, included in the manuscript, which we hope adequately conveys the message of our manuscript. Below, you will find a point-by-point response to each suggestion, according to the reviewer.

Reviewer 1

The authors searched some databases about the link of some microRNAs and Arterio-venous or Cavernous Malformations and summarized their findings in this manuscript. I am sure the authors put a lot of time and efforts in literature search and manuscript preparation. Even if most of the evidence are vague and the association between the microRNA expression and the malformation development is quite weak. The information is still useful for researchers in this field. One suggestion is that the authors should concentrate on the Arterio-venous and Cavernous Malformations. The discussion about the link of the microRNAs with other conditions should be limited to the extent that is relevant to the main diseases.

Response: We are grateful for your remarks and suggestion. The reason we have also detailed the findings of miRNAs in other cerebral pathologies is to provide a basis for potential future research linking these other diseases to BAVMs and CCMs, for a better understanding of these malformations. Therefore, we feel that by reducing some of the discussion regarding the roles these miRNAs play in pathologies distinct to vascular malformations may affect the significance of our review. We have, however, included a paragraph on the limitations you kindly pointed out, explaining that the evidence is currently vague and the associations are weak. We also explained a possible reason why these miRNAs have not been as extensively researched in BAVMs and CCMs, it being that these are already curable lesions that may not be as enticing to research as other incurable and malignant diseases. We hope that you understand our perspective and agree with our modifications. If, however, you prefer the discussion to be slightly abbreviated and more focused on these two lesions, we would be happy to oblige and would do our best in that regard.

Reviewer 2:

The authors proposed an outstanding review dealing with miRNA and brain vasculature malformations. The text is easily readable and the quality of the figures perfectly illustrates the review. No modifications needed, this review is acceptable in the present form.

Response: We sincerely thank you for your kind remarks. We hope that you will agree with the modifications made for the revision of our manuscript.

Again, we would like to thank the editor and the reviewers for their patience and suggestions.

Reviewer 2 Report

The authors proposed an outstanding review dealing with miRNA and brain vasculature malformations. The text is easily readable and the quality of the figures perfectly illustrates the review. No modifications needed, this review is acceptable in the present form.

Author Response

(The authors gave the same response as above.)
